Pure oxygen ventilation during general anaesthesia does not result in increased postoperative respiratory morbidity but decreases surgical site infection. An observational clinical study

von Bormann Benno 1 bvb@jodu.de
Suksompong Sirilak 1
Weiler Jürgen 2
Zander Rolf 3
1 Department of Anesthesiology, Siriraj Hospital, Mahidol-University , Bangkoknoi, Bangkok , Thailand
2 Anästhesie Team Nordrhein , Dinslaken , Germany
3 Department of Physiology, Johannes Gutenberg-University , Saarstraße, Mainz , Germany
Abdullah Jafri
Electronic publication date: 2014 Oct 9
Publication date: 2014
Volume: 2
Electronic Location ID: e613
Received 2014 Jul 13; Accepted 2014 Sep 16
Copyright: © 2014 von Bormann et al.
Copyright year: 2014
Copyright holder: von Bormann et al.
License: This is an open access article distributed under the terms of the Creative Commons Attribution License, which permits unrestricted use, distribution, reproduction and adaptation in any medium and for any purpose provided that it is properly attributed. For attribution, the original author(s), title, publication source (PeerJ) and either DOI or URL of the article must be cited.
License URL: https://creativecommons.org/licenses/by/4.0/

Keywords: General anesthesia, Low flow ventilation, Pure oxygen ventilation, Surgical site infection, Postoperative hypoxia

Funding: There was no funding for this work.

==============================
Background. Pure oxygen ventilation during anaesthesia is debatable, as it may lead to development of atelectasis. Rationale of the study was to demonstrate the harmlessness of ventilation with pure oxygen.

Methods. This is a single-centre, one-department observational trial. Prospectively collected routine-data of 76,784 patients undergoing general, gynaecological, orthopaedic, and vascular surgery during 1995–2009 were retrospectively analysed. Postoperative hypoxia, unplanned ICU-admission, surgical site infection (SSI), postoperative nausea and vomiting (PONV), and hospital mortality were continuously recorded. During 1996 the anaesthetic ventilation for all patients was changed from 30% oxygen plus 70% nitrous oxide to 100% oxygen in low-flow mode. Therefore, in order to minimize the potential of confounding due to a variety of treatments being used, we directly compared years 1995 (30% oxygen) and 1997 (100%), whereas the period 1998 to 2009 is simply described.

Results. Comparing 1995 to 1997 pure oxygen ventilation led to a decreased incidence of postoperative hypoxic events (4.3 to 3.0%; p < 0.0001) and hospital mortality (2.1 to 1.6%; p = 0.088) as well as SSI (8.0 to 5.0%; p < 0.0001) and PONV (21.6 to 17.5%; p < 0.0001). There was no effect on unplanned ICU-admission (1.1 to 0.9; p = 0.18).

Conclusions. The observed effects may be partly due to pure oxygen ventilation, abandonment of nitrous oxide, and application of low-flow anesthesia. Pure oxygen ventilation during general anaesthesia is harmless, as long as certain standards are adhered to. It makes anaesthesia simpler and safer and may reduce clinical morbidity, such as postoperative hypoxia and surgical site infection.

Introduction

Abandoning nitrous oxide for general anaesthesia enables the risk-free performance of low flow anaesthesia (Baum & Aitkenhead, 1995), and as a consequence the oxygen fraction during anaesthetic ventilation (FiO2) is an increasingly debated issue (Baum et al., 2004). Ventilating patients by using a closed circuit ensures that the lowest possible flow corresponds with the oxygen uptake of the individual; which is 200–300 ml/min for an adult (Zander, 1990). Applying consequent low flow ventilation has been the key reason for the paradigm change of our department as described in this study; abandoning nitrous oxide was a by-product of this strategy.

There is concern about high oxygen fraction deteriorating pulmonary function, as it is known to promote the development of atelectasis (Hedenstierna & Edmark, 2010). However, up to now there is no scientific evidence of this effect being relevant for general anaesthesia in otherwise adequately treated patients (Staehr et al., 2012).

The rationale of our study was to demonstrate the harmlessness of pure oxygen ventilation after realizing that it is still (1) controversially debated (Canet & Belda, 2011), (2) not recommended (Qadan & Akca, 2012) and (3) as far as we know nowhere else routinely applied. Our study, which may be the first of its kind, reports the experience of more than 13 years after switching from 30% oxygen plus 70% N2O ventilation to 100% oxygen.

Methods

Comprehensive data analysis was started after ethical approval was granted by the Siriraj Institutional Review Board (Si 202/2013; 158/2556/E4). The data were collected at the Catholic Clinic Duisburg, now Helios-Klinikum Duisburg, Germany; their use was granted by the General Manager, according to the letter dated October 22, 2012.

Patients: Patients with general, vascular, orthopaedic and gynaecological surgery under general anaesthesia between 01.01.1995 and 31.12.2009 except premature infants were included. There were no other exclusion criteria.

Study questions: First, does pure oxygen ventilation affect the incidence of clinically relevant postoperative respiratory problems (oxygenation) and unplanned admission to the Intensive Care Unit (ICU)? Additionally, does pure oxygen ventilation decrease surgical site infection (SSI) and does it influence postoperative nausea and vomiting (PONV); is there an effect on hospital mortality?

Change to pure oxygen and data collection

Until 1996 anaesthetic ventilation in our department comprised 70% nitrous oxide plus 30% oxygen (FiO2 = 0.3). In preparation for a fundamental change of the ventilation strategy, meticulous data collection as described below was started 1st January 1995, the last complete year with 30% oxygen ventilation. In January 1996 a testing phase was started with varying anaesthetic ventilation, 100% oxygen vs. 30% oxygen + 70% nitrous oxide. Upon completion, data analysis showed no negative clinical outcomes, particularly no respiratory problems after high oxygen ventilation. The final decision to switch to pure oxygen was made in August 1996. Through SOP (standard operating procedure) No 1.2.1/96 pure oxygen ventilation was mandated for all patients with general anaesthesia. Details of anaesthetic treatment during 1995 and from August 1996 to 2009 are summarized in Table 1. Figure 1 shows the original display of a ventilator during low flow pure oxygen ventilation (Primus™; Dräger AG, Lübeck, Germany) in 1997.

Figure 1 Respirator picture.

Original display of a Primus™ Ventilator (Dräger, Lübeck, Germany) during general anesthesia in low-flow pure oxygen mode with Desflurane™. The picture was made in 1997; the exact date is no longer available.

Table 1 Anesthetic patterns with alternative strategies.

Specifics of intraoperative anesthetic measures in 1995 and from August 1996 until 2009.

Anaesthesiological characteristics	1995	From 16th August 1996	
Ventilator	Dräger Primus™	
Preoxygenation (Flow, Time)	5 L, 5 Min	
Tidal volume (ml/kgbw)	6–8 (mild hypercarbia—pCO2 ≈ 45 mmHg)	
PEEP (cmH2O)	⩾5	
Inhalation anaesthetic	Desflurane; in Infants ⩽ 5 Y Sevoflurane	
Gas Monitoring: Anaesthetic,
O2, N2O, CO2	Inspiratory and expiratory (Required by law)	
Nitrous-Oxide (%)	70	0	
Flow (l/Min)	2.5–5	0.2–0.3	
FiO2	0.3	1.0	
Inspiratory Oxygen (%; approx.)
after equilibration period	28	90	
Expiratory Oxygen (%; approx.)
after equilibration period	25	86	
Wash-In Phase (Min)	5	0	
Wash-Out Phase (Min)	5	0	
Induction, Relaxation, Opiate	Propofol (Etomidate),
Rocuronium, Fentanyl	Propofol (Etomidate),
Rocuronium, Remifentanil	
Risk of low oxygen supply	Yes	No	
Readjustment	Yes	No	
Monitoring—Basic = standard	ECG, NIBP, Temperature (oesophageal),
Pulsoxymetry, Relaxometry (MSD, UK),
Cuff-Pressure	
Advanced monitoring, selective for:
Laparotomy (all disciplines), thoracotomy,
surgery of the arteries
(Aorta, carotid, subclavian, lower limb),
Spine surgery; high risk patients	Invasive BP, C.I. (invasive and non-invasive)
incl. various calculated hemodynamics,
Somatosensory evoked potentials (SEP)
Hourly blood gas analysis (Radiometer Medical,
Bronshoj Denmark),
incl. blood sugar, lactate, cHb, platelet count	
Adjuvant measures—all	Gastric tube, Thermal blanket—Baer Hugger™
(incl. PACU)	
Adjuvant measures —all female
—all patients with PONV history	—Dexamethasone 4 mg IV during induction
—additional Ondansetron 16 mg IV	
Advanced measures	Central venous catheter, pulmonary artery catheter	
Antibiotic prophylaxis (SOP: IL-Nr:1.4-1992)	According to the respective actual guidelines	

Data were recorded by the anaesthesiologists responsible, residents, staff members, senior consultants, and head of department respectively using adapted forms. Central documentation and maintenance of data was performed by the head of the department and one of the senior consultants (JW). Patients’ data were kept confidential until discharge or death, and then condensed and transferred into anonymous files without traceable personal characteristics. These files summarize parameters as described below.

Parameters and organization

The following parameters in summarized pattern were available for evaluation:

Patient’s age (0–15, 16–70, >70 years old), female gender, ASA risk classification (I + II, III, IV) following the modified score of Lutz & Peter (1973), regional pain catheter (yes/no), homologous and/or autologous blood transfusion (yes/no), operation time (⩽ 90, >90 min).

Surgical intervention—accurately defined groups (Table 2 and results).

Postoperative hypoxia (up to 24 h postoperatively) defined as O2-Sat <92% while breathing normal air without spontaneous recovery and the need for treatment, such as supplemental oxygen and/or CPAP assistance.

Unplanned ICU-admission during hospital stay. This group includes patients with severe pulmonary complications after surgery.

Surgical site infection (SSI), defined as wound infection during hospital stay with at least positive bacterial culture after smear test. The information was gained from surgeons and/or ward staff members on the occasion of daily contacts. Cooperation was undisturbed.

Postoperative nausea and/or vomiting (PONV; up to 24 h postoperatively) without differentiation between nausea and vomiting. It was estimated positive the patients judging it as ‘very unpleasant’.

Hospital mortality; no follow up after discharge.

All operations with general anaesthesia were performed in a central area with 7 ORs plus 7 + 7 connected areas for anaesthesia care. Directly connected to the ORs were the Postoperative Care Unit (PACU) with 6 beds and the ICU with 12 beds, both run by the anaesthetic department. Patients not scheduled for ICU were postoperatively moved to PACU (all). Oxygen 2–4 l/min was applied via nasal tube, which was continued on the ward until the 1st postoperative day and beyond at treating surgeon’s discretion. All patients were visited by ‘their’ anaesthesiologist on postoperative day one and again if appropriate. Crucial documentation, including the pre- and postoperative period, was under permanent scrutiny by senior staff members and the head of department. Every single protocol had to be signed by the department head or his proxy before filing.

Table 2 Characteristics of surgical procedures in 1995 and 1997.

Surgical discipline	Surgical procedure	1995 (FiO2 = 0.3)	1997 (FiO2 = 1.0)	
ALL		5,255	5,245	
General surgery	ALL	1,322	1,351	
Minor	765	838	
Major	231	220	
Colorectal	326	293	
Gynaecology	ALL	779	736	
Minor	510	471	
Major	189	190	
Mamma	80	75	
Orthopaedic surgery	ALL	1,769	1,749	
Minor	997	990	
Major	693	656	
Spine	79	103	
Vascular surgery	ALL	1,443	1,409	
Minor	342	350	
Aortic	271	244	
Major artery	630	620	
Cerebral artery	200	195	
Notes.

General surgeryMajor gastrectomy, Whipple operation, splenectomy, biliodigestive anastomosis, oesophagus resection, liver surgery, lung surgery

Colorectal surgery of colon, rectum, sigma

Minor herniotomy, strumectomy, cholecystectomy, appendectomy, biopsy and minor revisions

GynaecologyMajor laparotomy, abdominal hysterectomy, vulvectomy

Mamma all resections of the breast

Minor endoscopy, biopsy, curette, minor revisions

Orthopaedic surgeryMajor arthroplasty or major repair of knee, hip, shoulder, polytrauma of the skeleton

Minor osteosynthesis, removal of material, tendon repair, arthroscopy, nucleotomy, kyphoplasty, minor revisions

Spine fusion, vertebrectomy, laminectomy, spondylodesis

Vascular surgeryAortic open abdominal and thoracic repair

Cerebral carotid artery repair (95%), repair of subclavian or vertebral artery

Major artery revascularization of peripheral arteries, such as iliac, femoral, popliteal, incl. various bypass procedures, thigh amputation

Minor varectomy, invasive catheter insertion, pacemaker and port insertion, shunt surgery, minor revisions and amputations

Statistical analysis

Due to confounding factors data of the whole 15-year investigation period were not statistically evaluated but described in absolute numbers and/or percentages only. The well matching years 1995 (30% oxygen) and 1997 (100% oxygen) were compared applying statistical measures.

Comparing 1995 and 1997: Sample size calculation was based on surgical site infection rate in patients receiving 30% oxygen during general anaesthesia. According to the literature supplemental oxygen may lead to an approximate decrease of SSI between 25 and 50% (Greif et al., 2000; Bickel et al., 2011). It was assumed patients with 30% compared to 100% oxygen ventilation during general anaesthesia will have a 20% higher rate of infection. To detect a type I error of 0.05 and a type II error of 0.1 using nQuery Advisor 3.0 the required sample size in each group is 5,241.

Data were analysed using SPSS version 16.0 software (SPSS, Inc., Chicago, IL, USA). Categorical data such as sex, ASA physical status, prevalence of pain therapy, transfusion rate and incidence of side effects are presented as number (per cent) and compared using χ2 test. A p-value less than 0.05 was considered statistically significant.

Results

A total of 76,784 patients with general anaesthesia during 1995–2009 (15 years) are included, 66,226 of them (1997–2009) having received pure oxygen ventilation.

From 1995 to 2009

Age groups 0–15, 16–70, and >70 years: There was a slight increase of patients >70 in all groups, most pronounced in vascular surgery.

Gender: The ratio of female patients didn’t change significantly during the years, which was similar for the risk status (ASA score); only in gynaecological patients the rate of ASA III/IV-patients increased from below 20% in 1995, to nearly 40% in 2009.

Pain catheters: The rate of pain catheters applied in patients with general surgery was about 30% with little observed variation over the years; in gynaecology it was 6–10%, in orthopaedics and trauma 22–29%, and in vascular patients 23–27%.

Homologous blood transfusion (HT): The most frequent transfusions were red cells (OR, ICU, wards). The HT-rate was 16.7–25.4% in general surgery, 6.2–12.8% in gynaecology, 9.6–20.4% in orthopaedic surgery, and 12.6–18.4% in vascular surgery.

Autologous blood transfusion (AT): Most patients with AT were in vascular (up to 40.4%) and orthopaedic surgery (up to 34%), due to frequent autologous predeposit.

Operation time: In general, orthopaedic, and vascular surgery the relation ⩽ 1.5 h:>1.5 h was about 50:50–45:65 with no relevant changes during the years. In gynaecological patients the ratio was 58:52–70:30.

Outcome: Including all surgical patients postoperative hypoxia, surgical site infection (SSI) and postoperative nausea and vomiting (PONV) dropped significantly from 1995 to 1997 with a decreasing tendency during the years to follow, whereas unplanned ICU-admission and hospital-mortality remained to a large extent stable.

Figure 2 shows the outcome of all patients independent from surgical speciality regarding the observed five parameters. Including all 4 investigated surgical disciplines Figs. 3 and 4 show the incidence of postoperative hypoxia and the incidence of surgical site infection (SSI) respectively.

Figure 2 Course of all patients during 15 years.

Outcome parameters of patients (all) with general, gynecological, orthopedic and vascular surgery between 1995 and 2009. Pure oxygen ventilation was started in August 1996. SSI, surgical site infection; PONV, postoperative nausea and vomiting.

Figure 3 Oxygen ventilation and postoperative hypoxia.

Postoperative hypoxia in patients of four surgical disciplines (all patients) during 1995–2009. Pure oxygen ventilation was started in August 1996.

Figure 4 Oxygen ventilation and postoperative wound infection.

Surgical site infection (SSI) in patients of four surgical disciplines (all patients) during 1995–2009. Pure oxygen ventilation was started in August 1996.

Comparing 30% O2 + 70% N2O (1995) and 100% O2 (1997); Tables 2–4

Table 3 Summarized characteristics of surgical patients 1995 and 1997 in %.

Surgery	ALL	General	Gynaecology	Orthopaedic s/Trauma	Vascular	
Parameter	1995	1997	p	1995	1997	p	1995	1997	p	1995	1997	p	1995	1997	p	
N	5,313	5,245		1,322	1,351		779	736		1,769	1,749		1,433	1,409		
<15 Y	10.1	11.0	0.36	16.6	17.8	0.69	7.8	9.5	0.23	13.5	14.6	0.57	1.2	0.8	<0.0001	
16–70 Y	78.8	76.3	65.2	64.6	79.1	75.4		79.9	78.5	89.9	85.2	
>70 Y	11.1	12.7	18.2	17.6	13.1	15.1	6.6	6.9	8.9	14.0	
Female	47.0	48.5	0.18	47.1	49.1	0.16	100	100	1	37.2	39.3	0.19	30.2	32.5	0.2	
ASA I–II	42.1	40.3	0.33	51.1	51.0	0.41	82.9	82.7	0.81	46.1	41.1	0.12	7.0	6.9	0.9	
ASA III	32.8	33.6	24.2	22.4	13.0	12.5		46.9	51.1	33.7	33.6	
ASA IV	25.1	26.1	24.7	26.6	4.1	4.8	7.0	7.8	59.2	59.5	
Pain-Catheter	26.2	26.7	0.72	31.8	31.3	0.8	9.4	12.2	0.63	22.9	22.6	0.86	34.0	34.7	0.71	
H-Trans	17.2	16.0	0.60	22.5	21.3	0.68	6.2	6.3	0.68	20.4	18.8	0.68	14.6	12.6	0.3	
A-Trans	19.7	21.4	1.8	2.1	0.4	0.7	31.4	33.5	32.2	32.7	
OP ⩽ 1.5 h	48.7	50.0	0.9	55.4	58.8	0.68	66.0	63.0	0.85	47.5	49.0	0.35	36.7	35.7	0.9	
OP >1.5 h	51.3	50.0	44.6	41.2	34.0	37.0	52.5	51.0	63.3	64.3	
Notes.

Pain-Catheter regional catheters for postoperative pain relief (epidural, femoral, sciatic, interscalene)

H-/A-Trans homologous/autologous transfusion (any kind)

Table 4 Outcome data of surgical patients 1995 (30% oxygen) and 1997 (100% oxygen).

Surgical group	N	Postop. Hypoxia (%)	U-ICU (%)	SSI (%)	PONV (%)	H-Mortality (%)	
	1995	1997	1995	1997	P	1995	1997	P	1995	1997	P	1995	1997	P	1995	1997	P	
ALL	5,313	5,245	4.3	3.0	<0.0001	1.1	0.9	0.18	8.0	5.0	<0.0001	21.6	17.5	<0.0001	2.1	1.6	0.088	
General surgery	1,322	1,351	3.9	2.7	0.026	0.8	0.8	0.959	10.8	6.1	<0.0001	23.5	19.0	0.004	2.5	1.9	0.136	
Colorectal	326	293	3.4	2.7	0.643	0a	0a		22.4	14.7	0.014	20.6	17.7	0.377	4.3	4.1	0.902	
Major	231	220	3.5	2.3	0.514	0a	0a		17.3	10.9	0.051	25.6	20.9	0.245	6.9	4.1	0.110	
Minor	765	838	4.1	2.7	0.114	1.4	1.3		3.9	1.8	<0.01	24.2	19.0	0.011	0.4	0.5	0.796	
Gynecology	779	736	2.3	2.2	0.581	0.5	0.1	0.20	8.5	5.4	0.021	21.3	16.8	0.027	0.8	0.4	0.321	
Major	189	190	5.8	4.2	0.473	2.1	0.5	0.175	12.7	7.9	0.124	20.1	14.2	0.128	1.6	1.1	0.391	
Mastectomy	80	75	3.8	4.0	0.936	0	0		13.8	8.0	0.252	23.8	18.7	0.44	2.5	0	0.168	
Minor	510	471	0.8	0.6	0.784	0	0		6.1	4.0	0.146	21.4	17.6	0.139	0.2	0.2	0.792	
Orthopedics	1,769	1,749	4.5	2.6	0.003	0.6	0.4	0.357	2.1	1.5	0.224	20.6	16.5	0.002	0.5	0.4	0.632	
Spine	79	103	6.3	3.9	0.451	0a	0a		1.3	1.0	0.850	31.6	21.4	0.116	2.5	1.0	0.412	
Major, Arthroplasty	693	656	4.6	2.3	0.020	1.0	0.3	0.112	2.7	1.8	0.264	20.5	16.9	0.093	0.7	0.6	0.801	
Minor	997	990	4.3	2.7	0.055	0.4	0.2	0.418	1.7	1.4	0.601	20.2	15.8	0.011	0.2	0.2	0.994	
Vascular surgery	1,443	1,409	5.4	4.9	0.128	2.2	1.8	0.481	12.3	7.9	<0.0001	21.3	17.5	0.009	5.5	4.8	0.531	
Aorta	271	244	8.1	6.1	0.387	0a	0a		6.3	3.3	0.114	18.1	14.8	0.300	5.2	4.5	0.729	
Periph. Arteries	630	620	6.5	5.5	0.446	4.4	3.7	0.511	21.9	14.7	<0.001	25.1	20.0	0.053	7.3	6.1	0.405	
Cerebral Arteries	200	195	2.0	0.5	0.186	0a	0a		5.5	3.1	0.235			0.151	2.5	1.5	0.498	
Minor	342	350	3.2	2.6	0.613	1.2	1.0	0.681	3.5	2.0	0.225	17.0	13.4	0.196	0	0		
Notes.

a Patients generally designated for ICU.

U-ICU Unplanned ICU admission

SSI Surgical site (wound) infection

PONV postoperative nausea and vomiting

H-Mortality in-hospital mortality (during hospital stay; no follow up)

Group characteristics: With exception of increase of elderly (>70 years) from 8.9% (1995) to 14.0% (1997) in the vascular group, there were no significant differences between 1995 and 1997 regarding surgical procedures (Table 2), patient’s age, gender, ASA status, pain management, and transfusion requirement (Table 3).

Outcome: The incidences of the five recorded parameters are demonstrated in Table 4.

Postoperative hypoxia: Pure oxygen ventilation led to a decrease of postoperative hypoxia in all groups and subgroups, which was significant in all patients (4.3% to 3.0%; p < 0.0001), in patients with general surgery (all; p = 0.026), in orthopaedic patients (all; p = 0.003), and in arthroplasty patients (p = 0.020). There was no report of atelectasis related hypoxemia.

Unexpected ICU-admission (U-ICU): the frequency of U-ICU was generally low with no significant changes between 1995 and 1997.

Surgical site infection (SSI): Comparing all patients without dividing in surgical groups or subgroups SSI decreased from 8% in 1995 to 5% in 1997 (p < 0.0001), and decreased also within all subgroups which was statistically significant in general surgery, and in patients with peripheral artery surgery (Table 4).

Postoperative nausea and vomiting (PONV): Decrease of PONV was significant for all patients, all surgical disciplines in total, and some subgroups, such as minor general and minor orthopaedic surgery and patients with operations of the peripheral arteries.

Mortality: Overall hospital mortality dropped from 2.1% to 1.6% (p = 0.088). There were no significant differences within the subgroups.

Discussion

Findings: We found that high oxygen ratio leads to a decrease of postoperative hypoxia and overall in-hospital mortality as well as a reduction of surgical site infection and postoperative nausea and vomiting. There were no adverse effects.

Data quality and clinical standards: This single-centre study analysed follow-up routine data, which were steadily and prospectively collected in a well manageable and tightly organized anaesthetic division. Initially publication was not planned; therefore patient consent were not obtained. Because data generally belong to the respective hospitals, their use had to be granted by the general manager, which was done with letter from October 2012.

It is known that quality of anaesthetic management, such as preoperative antibiotics, maintenance of perioperative normothermia and optimized pain therapy relevantly affect patients’ outcome (Forbes & McLean, 2013). In our department, quality standards (SOP) have been implemented since 1989, which is long before acquisition of the presented data. Almost all of our patients, when indicated, had regional pain catheters. Transfusions, known to deteriorate surgical patients’ outcomes (Ferraris et al., 2012), were standardized by specific mandatory transfusion instructions (Schleinzer, Kasper & von Bormann, 1995). Red cells were given when cHb was <8.0 (homologous) or <9.0 g/dl (autologous) respectively. All patients, even for the shortest surgical procedure were provided with a warming blanket which included the postoperative period when necessary. The consistency of anaesthetic management is demonstrated in Table 1. The influence of the individual anaesthesiologist on outcome is considered to be limited. Tables 2 (surgical procedures) and 3 (group characteristics) show the comparability of the data 1995 vs. 1997, which is in accordance with the Statistical Department, Mahidol University.

Confounding factors: Within 15 years, surgical techniques and hygiene standards improved markedly, and so did equipment and performance in anaesthesia and intensive care. In 1997 the mean operation time for total hip arthroplasty in our hospital was 2 h 30 min, the average blood loss 1,000 ml and transfusion requirement (autologous and/or homologous) 100%. In 2000 with a new orthopaedic crew it was 45 min, 150 ml and 35% (almost entirely autologous) respectively. Between 1995 and 2009 the chairmen of all surgical disciplines included in this data collection changed. Only the head of anaesthesia (BvB) remained the same. However, assessing our data we have to include not only pure oxygen ventilation, strategy but also ventilating in low-flow instead of high-flow mode and using nitrous oxide plus oxygen or oxygen alone.

Low flow ventilation. Though the flow during anesthetic ventilation does not influence tidal volume or endexpiratory pressure, there is a significant effect on lung function and —integrity. Bilgi et al. (2011) in their clinical study compared 1 l/min ventilation during anesthesia with 3L/min in otherwise healthy individuals. They found that respiratory function and mucociliary clearance are better preserved after low-flow anesthesia. Humidity and temperature of the gas was more stable in low-flow than in high-flow anesthesia. The positive effects on postoperative lung function observed in our study may be partly due to the applied low-flow mode, an aspect inadequately represented within the literature about supplemental oxygen.

Abandoning nitrous oxide. We have to point out that we compared 100% oxygen vs. 30% oxygen plus 70% nitrous oxide. Therefore the impact of nitrous oxide and its absence since 1996 on our findings has to be considered. The influence of nitrous oxide on human metabolism and outcome parameters has been extensively investigated (Myles et al., 2006; Pasternak et al., 2009). Leslie et al. (2013) in the POISE trial demonstrated that nitrous oxide was not associated with adverse outcome such as myocardial infarction, stroke, infection, significant hypotension and death. Turan et al. (2013) even found a decreased risk of hospital morbidity and 30-day mortality in patients with N2O compared to others without, whereas Chen et al. (2013) reported N2O leading to damage of leucocyte DNA and an increased risk of infection in patients with colorectal surgery. However, nitrous oxide is a fast spreading gas moving in every cavity available such as pleura, bowel or endotracheal tube-cuff causing distension (Akca et al., 2004) an effect not wanted by anesthesiologists and surgeons. Against the background of current scientific evidence the influence of lacking nitrous oxide on our data is negligible, except the prevalence of postoperative nausea and vomiting (PONV).

Postoperative oxygenation, pulmonary morbidity: Changing the anaesthetic ventilation strategy to 100% oxygen in 1996, we knew that formation of atelectasis during general anaesthesia was reported to be more pronounced in patients breathing high oxygen concentrations (Rothen et al., 1995), but we had also read the studies of Lampron et al. (1985) and Lemaire et al. (1985), demonstrating no negative effect of pure oxygen ventilation in patients with respiratory failure or severe bacterial pneumonia. Additionally it is known that high oxygen ventilation prevents the lung from hypoxic pulmonary vasoconstriction by neutralizing the Euler-Liljestrand mechanism (Sommer et al., 2008), which may contribute to the beneficial effect, seen in our study. Today, eighteen years after we started with pure oxygen ventilation, atelectasis during anaesthesia is still not proven to relevantly affect postoperative outcome in otherwise adequately treated patients. In our study postoperative oxygenation was not negatively affected by pure oxygen ventilation and there was no increased incidence in postoperative pulmonary morbidity, which went for all groups and subgroups. Postoperative hypoxic events even decreased slightly. Our data can be compared with the clinical investigation of Mackintosh et al. (2012), who applied an intraoperative FiO2 of either 0.3 or 0.9 with and without PEEP and followed up the patients 24 h after extubation. There was no difference between the groups regarding postoperative oxygenation and the need for additional oxygen. There are also studies with sophisticated approaches, such as computer tomography (Akca et al., 1999), measurement of oxygenation index (PaO2/FiO2) and functional residual capacity (Staehr et al., 2012; Kanaya, Satoh & Kurosawa, 2013) and meta-analyses of randomized trials (Qadan et al., 2009; Hovaguimian et al., 2013) comparing patients being ventilated with 30%, 40%, 80% or 100% oxygen respectively. None of these studies found any deleterious effect of high oxygen ratio during anaesthetic ventilation. However, especially the conclusions of the meta-analysis of Hovaguimian et al. (2013) regarding the lack of pulmonary side effects induced a fierce controversial debate (Belda et al., 2014; Hedenstierna & Edmark, 2014; Hovaguimian et al., 2014; Meyhoff et al., 2014). Hedenstierna and co-workers referred to their own investigations (Hedenstierna & Edmark, 2014; Hedenstierna, 2012) and experimental data of van Kaam et al. (2004) showing that atelectasis increase the incidence of pneumonia. However, regarding the clinical relevance, they gave no answer. Meyhoff et al. (2014) criticized that in most studies routine pulmonary examinations have not been performed and, ‘adverse effects may be greatly underdiagnosed’. We too can only present parameters of ‘real life’ routine. Patients after surgery having a normal clinical course and being discharged after the usual length of hospital stay, do not undergo additional diagnostics without justified indication. Observing postoperative oxygenation in all our patients closely for 15 years, more than 13 years of which were with pure oxygen ventilation, we experienced no negative effect of supplemental oxygen at all, though we did not apply any prophylactic measures such as intraoperative recruitment manoeuvre. However, anaesthetic treatment was strictly adjusted, including intra- and postoperative body-temperature conservation, pain therapy, ventilation with PEEP and low tidal volume. Finally we want to point to a routine anaesthetic measure, relaxation, which is nowhere discussed in the literature regarding supplemental oxygen and lung function. Relaxation ‘by the clock’ is a known risk factor for respiratory complications (Grosse-Sundrup et al., 2012). As pointed to by Donati (2013), residual paralysis after general anesthesia has an incidence up to 57%, which is appreciated by only 1% of anesthesiologists. In our patients relaxation was restrictively applied under continuous monitoring (relaxometry).

Surgical site infection (SSI): It is known from experimental and clinical data that wound healing and integrity of gastrointestinal anastomosis significantly depends on tissue oxygenation (Knighton, Halliday & Hunt, 1984; Schietroma et al., 2013; Kotani et al., 2000). It is also known that nitrous oxide has no effect on SSI (Fleischmann et al., 2005). Comparing 30% to 100% oxygen ventilation within two well comparable groups with identical treatment regarding surgical technique, timing of antibiotics, anaesthetic care incl. temperature control and pain therapy (Tables 1–3), there was a SSI-reduction in all surgical disciplines and subgroups (Table 4). The highest SSI- and consequently reduction-rates occurred in colonic surgery, patients with major abdominal approach, and surgery of peripheral arteries. The incidence of SSI in colonic surgery is high with 20% (Yokoe et al., 2012) to 36% (Hubner et al., 2011). In their large randomized study in patients with colorectal resection and either 30% or 80% oxygen ventilation Greif et al. (2000) found supplemental oxygen reducing SSI from 11.2 to 5.2%, which is low compared to our findings in similar patients (22.4 to 14.7%). However, Greif et al. (2000) investigated randomized groups excluding patients with minor colon surgery as well as patients with history of fever and infection and patients with serious malnutrition. Our patients were consecutive, with a rate of ASA III/IV patients of 40.5/50.6% (1995) and 33.8/60.1% (1997) respectively, whereas Greif et al. (2000) did not include ASA IV patients at all and ASA III patients were 15–18%. Our data match with the findings of Belda et al. (2005). In their randomized controlled study in patients with elective colorectal surgery they found 24.4% SSI in patients with 30% and 14.9% in patients with 80% oxygen. In general the discussion about the benefit of high oxygen ratio on surgical site infection is still open, meta-analyses providing conflicting conclusions (Qadan et al., 2009; Togioka et al., 2012).

Postoperative nausea and vomiting (PONV): The scientific debate about the influence of high oxygen ventilation on PONV is controversial (Joris et al., 2003; Rincon & Valero, 2008). In our study we compared nitrous oxide (70%) with no nitrous oxide (100% oxygen). Leslie et al. (2008) in a randomized trial found N2O significantly led to an increase in PONV. Therefore the reduction of PONV in our study may be rather a consequence of ventilation without N2O than ventilating with pure oxygen.

Simplicity and safety: Anaesthesia with pure oxygen ventilation is simple when compared to any kind of air-gas- or nitrous oxide-gas mixture, because only the gas flow (approximately equivalent to patients’ oxygen consumption) has to be adjusted. Most importantly it provides maximum safety. For one thing, confusing oxygen with air or nitrous oxide is impossible. For another thing pulmonary oxygen storage, including residual volume and the physically dissolved oxygen, accommodates the maximum possible oxygen. Physical oxygen is important not by its amount but by its availability (Feiner et al., 2011) which can be crucial in case of cardiopulmonary problems or sudden blood loss with deteriorated circulation.

Study limitations: The assignment of particular data, such as specific co-morbidities, biometric and laboratory data, rate of emergency cases to the individual patient was no longer possible, restricting statistical analysis options. Thus the findings about morbidity parameters, such as SSI and PONV have to be interpreted with care. We present clinical observational data reflecting daily routine when sophisticated diagnostic procedures such as CT or lung function tests are applied only for a justifiable indication.

Conclusions

The data presented in this study support the thesis that pure oxygen low-flow ventilation during general anaesthesia is simple and provides a high degree of safety independent of the equipment of the individual department. It is harmless if therapists adhere to strict patient management standards, such as temperature control, optimized pain regimen, guideline-adapted antibiotic therapy and restrictive use of relaxants and allogeneic blood transfusions. Low-flow anesthesia seems to have a lung protective effect keeping humidity and temperature of the gas stable.

Supplemental Information

Supplemental Information 1 Postopertative hypoxia in %. All patients

Click here for additional data file.

Supplemental Information 2 Postoperastive hypoxia in numbers. All patients

Click here for additional data file.

Supplemental Information 3 Hospital mortality in %. All patients

Click here for additional data file.

Supplemental Information 4 Hospital mortality in numbers. All patients

Click here for additional data file.

Supplemental Information 5 Postoperative nausea and vomiting (PONV) in %. All patients

Click here for additional data file.

Supplemental Information 6 Postoperative nausea and vomiting (PONV) in numbers. All patients

Click here for additional data file.

Supplemental Information 7 Surgical site infection (SSI) in %. All patients

Click here for additional data file.

Supplemental Information 8 Surgical site infection (SSI) in numbers. All patients

Click here for additional data file.

Supplemental Information 9 Unplanned admission on intensive care unit (ICU) in %. All patients

Click here for additional data file.

Supplemental Information 10 Unplanned admission on intensive care unit (ICU) in numbers. All patients

Click here for additional data file.

Supplemental Information 11 Patients between 1995 to 2009 within four disciplines divided into groups regarding the extend of surgery (numbers and %)

Click here for additional data file.

Supplemental Information 12 Particular surgical procedures from 1995 to 2009 within four disciplines (numbers and %)

Click here for additional data file.

We thank Dr Daniel Bressington, Senior lecturer, Canterbury Christ Church University, UK, for proof-reading this manuscript and lecturer Prasert Sawasdiwipachai, Siriraj hospital for assistance with literature research and data assessment.

This study is cordially dedicated to the late Jan Baum, Professor of Anaesthesiology, Damme, Germany, one of the ‘fathers’ of low flow anaesthesia.

Additional Information and Declarations

Competing Interests

Author Contributions

The authors declare there are no competing interests.

Benno von Bormann conceived and designed the study, performed the study, analyzed the data, wrote the paper, prepared figures and/or tables, reviewed drafts of the paper.

Sirilak Suksompong analyzed the data, contributed reagents/materials/analysis tools, wrote the paper, prepared figures and/or tables, reviewed drafts of the paper.

Jürgen Weiler conceived and designed the study, performed the collection and management of the data.

Rolf Zander analyzed the data, reviewed drafts of the paper, specific considerations about oxygen physiology/toxicity.

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
