# Peer review of "Pure oxygen ventilation during general anaesthesia does not result in increased postoperative respiratory morbidity but decreases surgical site infection. An observational clinical study"

_PeerJ, doi:10.7717/peerj.613_

## Round 0.1 · original submission · Major Revisions

Dear Authors,

There seem to be major ethical and statistical issues in this manuscript that need to be sorted out.

Reviewer 1 declined to fully review the manuscript due to their concerns about the ethical approval. They said:

"There seems to be a problem with ethical approval of this study. Data were collected at the Catholic Clinic Duisburg in Germany. Data analysis was started after ethical approval by the Siriraj Institutional Review Board of Thailand. Use of data was granted by the General Manager of the Hospital in Germany by a letter. This approach is unacceptable in my opinion. Ethical approval has to be obtained by the Ethical Committee of the state of the hospital. Otherwise it would be possible to "pool" ethical approval all over the world by the ethical committees with the lowest standards. Furthermore a General Manager of a Hospital cannot decide whom to send data regardless of an ethical vote of the country responsible."

If it is indeed true that your ethical approval is incomplete then the manuscript would have to be rejected as PeerJ publishes articles that have undergone rigorous human ethical reviews at the highest level. Therefore I am giving you an opportunity to address these concerns or to obtain the correct ethical approval(s).

In addition, reviewer 2 has concerns about the statistics.

Reviewer 2 ·

Basic reporting

No comment

Experimental design

This is a very interesting study in a large, precise German database.
Results are sound, but I fully agree with the authors that “we compared 100% oxygen vs. 30% oxygen plus 70% nitrous oxide”. Accordingly, the beneficial effects could be explained not only by the more oxygen, but also by the lack of nitrous oxide or both. This fact should be included much earlier in the paper. Additionally, this study compares “high” flow (with N2O and 30% O2) vs. low flow (with 100% O2) anesthesia as well, so there are too many confounding factors which are not emphasized sufficiently.
Low tidal volume ventilation with mild permissive hypercarbia is a very important factor in lung protection which technique was used by the authors already during 1995! This should be discussed more extensively, as this fact may be responsible why hyperoxia did not deteriorate pulmonary functions.

Validity of the findings

Unfortunately no data presented on lung mechanics at least during surgery when anesthesia machine itself may provide few data like compliance, etc.
This is a human study but there are preclinical data on large animals (Bart E Crit Care 2008, Hauser B Crit Care 2009) and even review paper (Calzia E Crit Care 2010), where antibacterial and catecholamine-like effects were attributed to hyperoxia without affecting pulmonary mechanics at all in a clinically relevant large animal model of fecal peritonitis which may support some of the data from this paper.
p=0,088 is NOT a significant result, but maybe a strong tendency, therefore, it should by stated accordingly in the abstract as well as in the text.
Figure 1 nicely presents the (possible hyperoxia-related) improvement in postoperative hypoxia, but these are the only data from this later period. I think that other relevant data (like SSI, PONV, etc. during 1996-2009) are also possible candidates of figures.

Additional comments

I think that these nice data needs more demonstration and discussion regarding study "design". This study compares “high” flow (with N2O and 30% O2) vs. low flow (with 100% O2) anesthesia. It is a good idea to compare 1995 and 1997, but there are not enough data to present the second period (1996-2009).

---

## Round 0.2 · Minor Revisions

Dear Authors,

The issues as you can read in the comments are mainly ethical, specifically I draw your attention to the "Comments for the Author" paragraphs from Reviewer 2 which I encourage you to consider.

In addition to the comments of this reviewer, we specifically sought advice regarding the ethical concerns that have been raised by the reviewers (including seeking advice from people familiar with German ethical requirements). The advice we received indicated that provided you can positively confirm that the data has been de-identified at both institutions (in both Germany and Thailand) then, together with the positive confirmation from the German institute, the ethical status of your study is appropriate. We appreciate this contradicts the prior expressed reservations of Reviewer 1, however this is the advice that we have received.

Please can you clarify these issues / confirm these requirements before we go further. Thank you for your attention to this matter.

Reviewer 2 ·

Basic reporting

The manuscript improved a lot (incl, more information on low flow anesthesia and lack of N2O), however, it is still a "pure oxygen ventilation" paper in the title...

Experimental design

This is a retrospective chart review on a prospectively collected database with missing data etc. This should be emphasized. No other comments

Validity of the findings

The discussion improved a lot, limitations are considered in depth.

Additional comments

The ethical issue raised by the other reviewer is valid. For me the real quiestion is: who owns the data?
It is clear that patient consent are not needed for this type of data collection (for quality insurance purposes) and is not possible after 15 years as well, but for publication? My opininon is that currently these data belongs to the German Insitute, and they may allow the data to be analysed by a "third party", the current authors. But still these data belongs to the Institute, therefore the Institue's written permission should be very clear or the institute's representative should appear among the authors. Perhaps the first author would like to mention his original institute (Catholic Clinic Duisburg) among the affiliations?
Or the letter of permission may appear as a supplementary file?

---

## Round 0.3 · accepted · Accept

Dear Authors,

This manuscript is accepted as the revisions required have been done according to the comments of the reviewers and the ethical standards required by PeerJ.

Reviewer 2 ·

Basic reporting

no comments

Experimental design

no comments

Validity of the findings

no comments

Additional comments

no comments